# Epigenetic Silencing of miR-33b Promotes Peritoneal Metastases of Ovarian Cancer by Modulating the TAK1/FASN/CPT1A/NF-κB Axis

**DOI:** 10.3390/cancers13194795

**Published:** 2021-09-24

**Authors:** Xueyu Wang, Mingo M. H. Yung, Rakesh Sharma, Fushun Chen, Ying-Tung Poon, Wai-Yip Lam, Benjamin Li, Hextan Y. S. Ngan, Karen K. L. Chan, David W. Chan

**Affiliations:** 1Department of Obstetrics & Gynaecology, Li Ka Shing Faculty of Medicine, The University of Hong Kong, Hong Kong, China; u3004659@connect.hku.hk (X.W.); h1094157@connect.hku.hk (M.M.H.Y.); chenfs@hku.hk (F.C.); michpoon@connect.hku.hk (Y.-T.P.); hysngan@hku.hk (H.Y.S.N.); 2Centre for PanorOmic Sciences Proteomics and Metabolomics Core, Li Ka Shing Faculty of Medicine, The University of Hong Kong, Hong Kong, China; rakesh.cpos@hku.hk; 3Lee’s Pharmaceutical (HK) Ltd., 1/F Building 20E, Phase 3, Hong Kong Science Park, Shatin, Hong Kong, China; wy.lam@leespharm.com (W.-Y.L.); drli@leespharm.com (B.L.)

**Keywords:** ovarian cancer, omental metastases, miR-33b, DNA methylation, lipid metabolism

## Abstract

**Simple Summary:**

Omental metastasis and peritoneal dissemination are frequently observed in ovarian cancer peritoneal metastases and are associated with high mortality and poor prognosis. The tumor microenvironment is known to influence cancer epigenomics, which plays an essential role in promoting tumor development and metastatic progression. Therefore, investigation of the epigenetic mechanisms underlying the growth of ovarian cancer cells in the omental metastatic microenvironment is of great importance. Here, we report that miR-33b is significantly silenced by DNA hypermethylation in metastatic ovarian cancer cells to adapt to a lipid-rich microenvironment. Restoration of miR-33b was shown to impair lipid metabolic activities and reduce the oncogenic properties of ovarian cancer cells by negatively regulating the TAK1/FASN/CPT1A/NF-κB pathway, indicating that targeting this signaling cascade may be a molecular therapeutic choice for ovarian cancer metastatic progression.

**Abstract:**

Peritoneal metastases are frequently found in high-grade serous carcinoma (HGSOC) patients and are commonly associated with a poor prognosis. The tumor microenvironment (TME) is a complex milieu that plays a critical role in epigenetic alterations driving tumor development and metastatic progression. However, the impact of epigenetic alterations on metastatic ovarian cancer cells in the harsh peritoneal microenvironment remains incompletely understood. Here, we identified that miR-33b is frequently silenced by promoter hypermethylation in HGSOC cells derived from metastatic omental tumor tissues. Enforced expression of miR-33b abrogates the oncogenic properties of ovarian cancer cells cocultured in omental conditioned medium (OCM), which mimics the ascites microenvironment, and in vivo tumor growth. Of note, restoration of miR-33b inhibited OCM-upregulated de novo lipogenesis and fatty acid β-oxidation in ovarian cancer cells, indicating that miR-33b may play a novel tumor suppressor role in the lipid-mediated oncogenic properties of metastatic ovarian cancer cells found in the omentum. Mechanistic studies demonstrated that miR-33b directly targets transforming growth factor beta-activated kinase 1 (TAK1), thereby suppressing the activities of fatty acid synthase (FASN) and carnitine palmitoyltransferase 1A (CPT1A) in modulating lipid metabolic activities and simultaneously inhibiting the phosphorylation of NF-κB signaling to govern the oncogenic behaviors of ovarian cancer cells. Thus, our data suggest that a lipid-rich microenvironment may cause epigenetic silencing of miR-33b, which negatively modulates ovarian cancer peritoneal metastases, at least in part, by suppressing TAK1/FASN/CPT1A/NF-κB signaling.

## 1. Introduction

Ovarian cancer is one of the most common and lethal cancer types in women worldwide [1]. Similar to other human solid cancers, metastases are a critical factor leading to death and recurrence in ovarian cancer patients [2,3]. Epithelial ovarian cancer (EOC) accounts for over 90% of ovarian cancer cases, and the deadliest and most frequent histological subtype is high-grade serous carcinoma (HGSOC), representing the largest subgroup (>70%) of EOC [2,3]. Due to the lack of early clinical symptoms, HGSOC often presents with “silent symptoms,” and the majority of cases are diagnosed at an advanced stage, accompanied by peritoneal metastases, which contribute to most ovarian cancer-related deaths and a typically abysmal prognosis [1,4,5]. Standard first-line interventions to treat advanced EOC are surgical debulking followed by cytotoxic platinum and taxane chemotherapy [6]. However, diffuse peritoneal dissemination of EOC, as well as a favorable microenvironment for tumor development and progression, contributes to acquired platinum resistance and disease recurrence in ovarian cancer patients [7,8]. Therefore, a better understanding of the pathophysiology of ovarian cancer is urgently needed to explore alternative and novel antitumor therapies.

Mounting evidence has suggested that cancer arises from the accumulation of genetic mutations and epigenetic alternations [9]. Although genetic changes are associated with neoplastic transformation and tumor development, the role of genetic variations in metastatic tumor capabilities has not been conclusively determined [10]. Recent studies have indicated minimal genetic divergence between primary and metastatic tumor sites in some contexts of human tumors [10]. Metastatic tumors originate from the primary tumor, and it is not surprising that most of the genomic features can be retained. For instance, Liu G et al. reported that metastatic tumor tissues retain most genomic signatures observed in the primary tumors in 15 cancer types [11]. Additionally, the comparative genomic analysis showed that primary and metastatic tumors possess common copy number alterations (CNAs) and mutational landscapes [12]. These findings suggest that genetic alterations may not be the dominant drivers for most solid tumors during their metastatic progression [10]. Unlike genetic alterations, epigenetic changes are relatively dynamic and reversible in response to environmental changes [13]. Increasing evidence has suggested that metastatic traits may be caused by epigenetic changes that enhance oncogenic properties during tumor progression [14]. Indeed, recent evidence has shown that epigenetic changes play a pivotal role in multiple metastasis steps by regulating critical genes involved in tumor metastasis [14]. Reyes et al. found that hallmark methylation changes in HGSOC is global DNA hypomethylation status and specific hypermethylation of tumor suppressor gene promoters, contributing to EOC invasion and metastasis [15,16]. In addition, Hentze et al. reported that differences in DNA methylation patterns were associated with the process of tumor metastasis [17]. However, there is still a lack of examples reporting epigenetic modifications in peritoneal metastases of EOC.

Stressful factors in the tumor microenvironment (TME), such as hypoxia and nutrient starvation, can induce epigenetic reprogramming in tumor cells and contribute to cancer drug resistance and disease progression [18,19]. DNA methylation is the major epigenetic mechanism modulating the expression of a variety of crucial tumor suppressors, such as BRCA1 and RASSF1A, in ovarian cancer [16]. Aberrant epigenetic mechanisms in cancer metastasis also include transcriptional control of microRNAs (miRNAs) via DNA methylation and histone modifications [20]. Metastatic ovarian cancer cells preferentially disseminate and colonize the omentum during peritoneal metastases of ovarian cancer [21]. Clinical evidence has suggested that ascites or the omentum act as a reservoir providing plenty of lipids, adipokines, and metabolic substrates that function as crucial drivers of epigenetic alterations in promoting ovarian cancer development and metastatic progression [5,22]. However, the mechanisms of epigenetic crosstalk between the ascites microenvironment and ovarian cancer cells remain unclear.

Given that the TME plays an important role in shaping the DNA methylation patterns of cancer cells and miRNA expression in cancer metastasis, we investigated epigenetic events, especially the expression of miRNAs, in the metastatic progression of EOC. To this end, methylome analysis was employed to examine methylation alterations in primary and metastatic omental tumor tissues. In this study, we identified that miR-33b is hypermethylated in metastatic ovarian tumor tissues in the omentum. Restoration of miR-33b negatively regulated the TAK1/fatty acid synthase (FASN)/carnitine palmitoyltransferase 1A (CPT1A)/NF-κB signaling axis, leading to suppression of peritoneal metastases of ovarian cancer. This indicates that targeting the miR-33b-mediated pathway may be a practical therapeutic approach to prevent and manage peritoneal metastases in ovarian cancer.

## 2. Materials and Methods

### 2.1. Cell Culture and Human Clinical Samples

A human embryonic kidney cell line (HEK293), high-grade serous subtype ovarian cancer cell lines (MES-OV and OV-90), and a clear cell subtype ovarian cancer cell line (ES-2) were purchased from American Type Culture Collection (ATCC, Manassas, VA, USA). High-grade serous subtype ovarian cancer cell lines (PEO1 and PEO4) were purchased from the European Collection of Authenticated Cell Cultures (ECACC). Human immortalized ovarian epithelial cell lines (HOSE 96-9-18, HOSE 11-12, and HOSE 6-3) and a high-grade serous subtype ovarian cancer cell line (OVCA433) were kindly provided by George Tsao (The University of Hong Kong, Hong Kong, China). A high-grade serous subtype (HGSC) ovarian cancer cell line (HEY) was kindly provided by Alice Wong (The University of Hong Kong). High-grade serous subtype (COV318) and mucinous cell subtype (COV644) ovarian cancer cell lines were kindly provided by Benjamin Tsang (University of Ottawa, Ottawa, ON, Canada). HOSE cell lines and OV-90 cells were cultured in a 1:1 mixture of Medium 199 and Medium 105; ES-2, OVCA433, COV318, COV644, and HEK293 cell lines were grown in Dulbecco’s modified Eagle’s medium; the MES-OV cell line was maintained in McCoy’s 5a modified medium; and the HEY and PEO1/4 cell lines were cultured in RPMI-1640 medium. All the cell lines were cultured in an incubator (5% CO_2_, 37 °C) with culture medium supplemented with 10% heat-inactivated fetal bovine serum (FBS) and 100 IU/mL penicillin/streptomycin. Clinical primary ovarian tumors, omental tissue samples, and ascites/lavage clinical samples were freshly isolated from ovarian cancer patients of Queen Mary Hospital with prior approval of the Institutional Review Board of the University of Hong Kong/Hospital Authority Hong Kong West Cluster (HKU/HA HKW IRS) (IRS Reference Number: UW 11-298).

### 2.2. Whole-Genome Methylation Profiling

Genomic DNA was isolated from six pairs of primary and metastatic ovarian tumor tissues, and sodium bisulfite was converted prior to prepping of the tissues for analysis on an Infinium MethylationEPIC beadChips 850K array (Illumina, San Diego, CA, USA) according to the Infinium HD assay methylation protocol guide. The whole-genome methylation study was performed by the Centre for Genomic Sciences, The University of Hong Kong. Subsequently, data files were subjected to QC via GenomeStudio Version 1.8, and statistical analysis was conducted using the R package ChAMP. CpG sites with FDR-adjusted *p* values ≤ 0.05 and delta beta values ≥ 0.2 were considered statistically significant.

### 2.3. Methylation-Specific PCR (MS-PCR) and Pyrosequencing Analysis

Genomic DNA was isolated from cell lines or clinical tissues using a NucleoSpin^®^ DNA Rapid Lyse kit (Takara Biochemicals, Tokyo, Japan). Next, bisulfite conversion was conducted using an EpiJET Bisulfite Conversion Kit (Thermo Scientific, Waltham, MA, USA) following the suggestions provided by the manufacturer. To carry out MS-PCR on cells, Methyl Primer Express™ Software v1.0 (Applied Biosystems, Inc. Foster City, CA, USA) was employed to design methylation-specific (MSP) and unmethylation-specific (USP) primers. PCR products were amplified using PrimeSTAR^®^ Max DNA Polymerase (Takara). To perform pyrosequencing analysis, PyroMark Q24 Software (Qiagen, Hilden, Germany) was utilized to design primers, and PCR products were submitted to the Centre for PanorOmic Sciences, The University of Hong Kong for the following pyrosequencing analysis. Primers for the methylation study are described in Appendix A.

### 2.4. Construction of Cell Lines with Stable miR-33b Expression

To establish stable polyclonal miR-33b-overexpressing cell lines, ovarian cancer cells were transfected with the pCMV-miR33b plasmid (MI0003646) using Lipofectamine 3000 reagent (Invitrogen, Waltham, MA, USA). After 48 h of transfection, ovarian cancer cells were sorted by GFP signal using a BD FACSMelody Cell Sorter (Faculty Core Facility, The University Hong Kong). After neomycin selection for 3–4 weeks, GFP-positive ovarian cancer cells were validated for miR-33b overexpression via real-time quantitative PCR.

### 2.5. Omental Conditioned Medium (OCM) and Commercial Kits

OCM was prepared as previously described [23]. Omentum was freshly collected, washed with PBS, and minced into small pieces. The omental mixture was subsequently added to the cell culture medium supplemented with 1% FBS for 24 h. OCM was filtered and stored at 4 °C prior to the removal of omentum tissues by centrifugation. To selectively remove lipids and cell debris, Cleanascite™ Lipid Removal Reagent (Biotech Support Group, Monmouth Junction, NJ, USA) was employed for OCM according to the manufacturer’s suggestions. Additional commercial kits and drugs for the detection of cellular lipid metabolism are summarized in Appendix A.

### 2.6. Real-Time Quantitative PCR

Total RNA was isolated from cells and tissue samples using TRIzolTM Reagent (Thermo Scientific) according to standard protocols. The concentration of the obtained RNA sample was measured with a Nanodrop 2000c spectrophotometer. Afterward, an equal amount of RNA was retrotranscribed to cDNA using a TaqMan™.

MicroRNA Reverse Transcription Kit (Thermo Scientific). The miRNA expression level was detected via real-time qPCR with the use of TaqMan^®^ MicroRNA Assays in a ViiA 7 Real-Time PCR System (Applied Biosystems, CA, USA). U6 snRNA was utilized as an internal control for normalization, and the relative expression of miR-33b was calculated based on the 2^−ΔΔCt^ method.

### 2.7. Western Blot Analysis

Protein lysates were prepared in cell lysis buffer (Cell Signaling Technology, Denver, CO, USA) and separated via sodium dodecyl sulfate-polyacrylamide gel electrophoresis (SDS–PAGE) before being transferred to FL-PVDF membranes. Blots were incubated overnight at 4 °C using primary antibodies (Appendix A). Next, the membrane was washed with TBST and subjected to fluorescence-conjugated secondary antibodies (LI-COR IRDye 680/800 CW, 1:15,000 dilution). Fluorescence visualization was performed using an LI-COR Odyssey CLx Imager (Lincoln, NE, USA).

### 2.8. Cell Proliferation, Migration and Invasion Assays

Cell proliferation was detected using an XTT cell proliferation kit (Roche, Basel, Switzerland). In short, cells (1 × 10^3^) were seeded in a 96-well plate and cultured with different treatments. A mixture of XTT reagents, which includes PBS, XTT labeling reagents, and electron coupling reagents, was added to each well and cultured with cells at 37 °C for 4 h. Finally, the absorbance at 492 nm was read, and relative cell viability was expressed as a fold change over the mean of the first day. Cell migration and invasion abilities were determined using Transwell cell migration assay kits (Corning, New York, NY, USA) and BioCoat™ Matrigel^®^ Invasion Chambers (Corning), respectively. Briefly, cell samples (1 × 10^5^ cells) were suspended in serum-free medium and seeded in 24-well transwell chambers. Different media containing 1% FBS were added to the lower chamber. After incubation, the migrated/invaded cells were stained and counted using microscopy.

### 2.9. CRISPR/Cas9-Mediated Gene Knockout

Knockout of the TAK1 genes was performed via transfection with a PX458 (SpCas9-2A-GFP) plasmid carrying sgRNA oligonucleotides of TAK1 (designed by E-CRISP http://www.e-crisp.org/E-CRISP/, accessed on 19 June 2021). After transfection, GFP-positive cells were sorted using a BD FACSMelody Cell Sorter, and the knockout effects were validated via western blot analysis.

### 2.10. Dual-Luciferase Reporter Assay

For the dual-luciferase reporter assay, luciferase constructs were generated by cloning the 3’ UTR sequences containing the potential miR-33b binding sites of TAK1 (wild-type/mutants) into a pmirGLO plasmid (Promega, Madison, WI, USA). HEK293 cells were seeded in 48-well plates and reached 80–90% confluence after 24 h of incubation. The cells were then cotransfected with the pmirGLO-3’UTR plasmid or vector control using LipofectamineTM 3000 (Invitrogen) for 48 h. Finally, luciferase activities were measured using a Dual-Luciferase Assay Kit (Promega) according to the manufacturer’s instructions.

### 2.11. In Situ Hybridization (ISH) and Immunohistochemistry (IHC)

ISH and IHC were performed as previously described [24]. For ISH, a miRCURY LNA miRNA ISH Optimization Kit (FFPE) 5 (Qiagen) was applied to detect the expression of miR-33b following the manufacturer’s instructions. For IHC, sections of clinical tissues (4-μm thick) and ovarian cancer tissue arrays (OV812, US Biomax, Rockville, MD, USA) were deparaffinized, hydrated, and treated with primary anti-TAK1 antibody (Abcam, Cambridge, MA, USA) at a 1:100 dilution. Images were analyzed using an Aperio ScanScope System (Department of Pathology, The University of Hong Kong).

### 2.12. Proteomics and Bioinformatics Analysis

Total protein was isolated from scrambled control- or stable miR-33b-expressing cells with 24 h OCM coculture and subjected to LC–MS/MS analysis using an Orbitrap Fusion Lumos mass spectrometer interfaced with Dionex 3000RSLC nanoLC according to the manufacturer’s protocol. To analyze MS data, MaxQuant proteomics software (version 1.5.3.30, Max Planck Institute of Biochemistry, Martinsried, Germany) was applied, and the UniProt Human protein database was used for peptide identification. The appropriate parameter setting for false discovery rate (FDR) calculation was FDR ≤ 1% at the peptide and protein levels. Downregulated targets were selected for the protein function clusters and gene set enrichment analysis using the KEGG pathway and Reactome pathway. Data visualization of biological pathways was generated by Hiplot (https://hiplot.com.cn/, accessed on 21 April 2021).

### 2.13. Integrative Genomic Analyses for The Cancer Genome Atlas Ovarian Serous Cystadenocarcinoma (TCGA-OV) Data

HTSeq count RNA sequencing data of TAK1 along with clinical information were downloaded from the TCGA-OV dataset (https://portal.gdc.cancer.gov/, accessed on 19 June 2021. Patients were then stratified into two groups, lower (0–50%) or higher (50–100%) TAK1 expression groups, on the basis of RNA sequencing data. For differential analysis, the R package DESeq2 was applied to identify differentially expressed genes between the two groups. For correlation analysis, the R stats package (version 3.6.3, https://stat.ethz.ch/R-manual/R-devel/library/stats/html/stats-package.html and accessed on 6 September 2021) was employed to explore candidate genes correlated with TAK1 expression.

### 2.14. In Vivo Intraperitoneal Dissemination Mouse Model

The animal study was conducted according to criteria outlined by The Committee on the Use of Live Animals in Teaching and Research of The University of Hong Kong (CULATR number: 5717-21). Using procedures similar to those described in our previous in vivo intraperitoneal mouse study [25], 2 × 10^6^ GFP-labeled ES-2 ovarian cancer cells were intraperitoneally injected (i.p.) into 6–8-week-old SCID female mice in groups of five. With the formation of palpable tumors, animals were sacrificed and imaged using fluorescence stereomicroscopy (Nikon, Tokyo, Japan). The body weight, ascites volume, tumor weight, and fluorescence and bright-field images of tumor nodules spreading in the murine intraperitoneal cavity were recorded. Metastatic tissues were fixed for subsequent IHC staining.

### 2.15. Statistical Analysis

Data are expressed as the mean value ± SEM of at least three independent experiments. Statistical analysis was performed via Student’s *t*-test or one-way ANOVA using GraphPad Prism 8 software (GraphPad Software, La Jolla, CA, USA). Correlation coefficients were computed according to the Spearman method. A *p*-value of <0.05 was considered statistically significant.

## 3. Results

### 3.1. miR-33b Is Frequently Silenced by DNA Hypermethylation in Metastatic Ovarian Cancer Cells

Given that epigenetic alterations are associated with alterations in gene expression and drive metastatic progression [26], it is interesting to identify potential tumor suppressor genes altered by DNA methylation in the metastatic tumor tissues of ovarian cancer peritoneal metastases. By comparing the methylation signatures of the primary tumor tissues and their corresponding omental metastatic tumor tissues (*n* = 6 pairs) using a methylation 850 chip assay according to our methylome analysis [27], we observed a remarkable increase in global DNA hypermethylation levels in metastatic omental tumors compared to primary ovarian tumors using GenomeStudio software (Version 1.8, Illumina, San Diego, CA)) (Appendix A). Among the highly significant differentially methylated position (DMP)-associated genes, we identified that the miR-33b gene is hypermethylated and may be involved in the dissemination of ovarian cancer cells in the omentum (Figure 1A). Indeed, miR-33b has been reported to be participated in the regulation of fatty acid metabolism [28]. Given that elevated lipid metabolic activities have been observed in ovarian cancer during peritoneal disseminated metastases [23], it is of interest to investigate the expression and functional roles of miR-33b in ascites-derived ovarian cancer cells. Hence, through qPCR analysis, miR-33b was shown to be commonly underexpressed in a panel of advanced ovarian cancer cell lines (HEY, ES-2, OVCA433, COV318, COV644, PEO1, PEO4, MES-OV, and OV-90) but not in normal HOSE cells (Figure 1B). Moreover, miR-33b was expressed at a relatively lower level in metastatic tumor tissues than in matched primary ovarian tumors (*n* = 18 pairs), indicating that the downregulation of miR-33b in metastatic ovarian cancer cells in the omentum may have clinical relevance (Figure 1C; ** *p* < 0.01).

To determine whether DNA hypermethylation is involved in the downregulation of miR-33b expression, the demethylating agent 5-aza-dC was applied to treat ovarian cancer cells. qPCR analysis revealed that 5-aza-dC (1 and 5 μM) treatment dose-dependently restored the expression of miR-33b in ES-2 and OVCA433 cells and elevated miR-33b expression by 8-fold in MES-OV cells (Figure 1D), indicating that DNA methylation occurred on the promoter of miR-33b. As expected, methylation-specific PCR (MS-PCR) analysis clearly confirmed that hypermethylation of the miR-33b promoter is the major cause of the downregulation of miR-33b in ovarian cancer cells (Figure 1E). Moreover, pyrosequencing analysis on 18 out of 22 paired specimens illustrated that DNA methylation is markedly enriched on the promoter of miR-33b in metastatic ovarian tumors compared with that of the matched primary tissues (*n* = 22 pairs) (Figure 1F; *** *p* < 0.001). Notably, 10 pairs of the hypermethylated specimens were overlapped with the paired samples tested above by qPCR analysis that 8 pairs of them consistently exhibited significantly lower the level of miR-33b in the metastatic ovarian tumors. These data collectively suggest that DNA methylation is highly correlated with the downregulation of miR-33b in metastatic ovarian cancer.

### 3.2. Overexpression of miR-33b Attenuates OCM-Mediated Oncogenic Properties of Ovarian Cancer Cells

The intraperitoneal tumor microenvironment supports omental tropism of ovarian cancer, and OCM has been shown to highly mimic the TME of ascites or the omentum in promoting ovarian cancer cell growth and metastasis [5,23]. To understand the functional role of miR-33b, we established polyclonal miR-33b-overexpressing ovarian cancer cell lines, ES-2 and MES-OV (Figure 2A). XTT cell viability assays showed increased cell proliferation of both the ES-2 and MES-OV cell lines on Day 4 when cocultured in OCM compared to the control medium, whereas OCM-promoted cell proliferation was inhibited by overexpression of miR-33b (Figure 2B). We and others have reported that ovarian cancer cells utilize free fatty acids in the ascites for energy production and cell synthesis [29]. To examine whether fatty acids in OCM are the main source of energy for tumors, all fatty acids in OCM were first removed by Cleanascite^TM^ Lipid Removal Reagent. XTT cell viability analysis was performed and showed that the cell growth rate of ES-2 and MES-OV cells was remarkably reduced when cocultured in lipid-depleted OCM (Figure 2B). Likewise, Transwell assays revealed that OCM-cocultured ovarian cancer cells showed up to a 5-fold increase in cell migration and invasion abilities (Figure 2C,D). However, both miR-33b overexpression and depletion of fatty acids by Cleanascite in OCM significantly impaired ovarian cancer cell migration and invasion (Figure 2C,D). Hence, our data suggest that OCM can support tumor growth and aggressiveness by providing fatty acids as an energy source and that miR-33b impedes the proliferation, migration, and invasion of ovarian cancer cells driven by OCM coculture.

### 3.3. Enforced miR-33b Expression Inhibited OCM-Mediated High Lipid Metabolic Activities in Ovarian Cancer Cells

To investigate pivotal biological pathways or mechanisms involved in the tumor-suppressive effects of miR-33b on OCM-cultured ovarian cancer cells, we performed a label-free quantitative liquid chromatography-tandem mass spectrometry (LC–MS/MS)-based proteomics strategy. Proteomic results illustrated that certain critical genes regulated by miR-33b are associated with metabolic activities, such as FASN, ATP1B1 and HADHB (Figure 3A). Gene set enrichment analysis (GSEA) analysis with the Reactome database revealed a significant decrease in the enrichment of genes associated with the metabolism of lipids in miR-33b-overexpressing ES-2 cells (Figure 3B). For enrichment analysis of KEGG pathways, we selected 92 genes downregulated by miR-33b across all biological replications and found that most of the genes targeted by miR-33b were enriched in metabolic pathways involving fatty acid metabolism (Figure 3C). Mounting evidence suggests that ovarian cancer cells utilize free fatty acids from the omental microenvironment for de novo lipogenesis and ATP production to support tumor growth [23]. Hence, the present study examined whether miR-33b altered the lipid metabolism of ovarian cancer cells in OCM and consequently led to tumor suppression. Consistent with previous findings, a lipid staining assay showed that OCM promoted the formation of lipid droplets and restoration of fatty acids in the cytosol of ovarian cancer cells. However, overexpression of miR-33b attenuated OCM-induced accumulation of lipid droplets in the cytoplasmic compartment of ES-2 and MES-OV cells (Figure 3D). Triglycerides have been recognized as bioenergetic fuels for cancer growth and exert a tumorigenesis-promoting effect [30]. Therefore, we evaluated triglycerides, and the results indicated that OCM profoundly enhanced the triglyceride levels of ES-2 and MES-OV cells by 1.5-fold and 1.4-fold, respectively, compared with the scrambled controls with normal medium. Conversely, the OCM-increased triglyceride level was reduced to 1.2–1.5-fold when ovarian cancer cells overexpressed miR-33b (Figure 3E). Lipolysis plays an essential role in ATP production and has protective effects on ovarian cancer cells [23]. A lipolysis colorimetric assay suggested that an 80–90% increase in lipolysis activity in ES-2 and MES-OV cells compared with scrambled controls was caused by OCM treatment, whereas such OCM-upregulated lipolysis activity was diminished by overexpression of miR-33b (Figure 3F). Cancer cells undergo β-oxidation to recruit fatty acids as an energy source and protect tumor cells from environmental stresses [31]. Here, OCM-cocultured ES-2 cells exhibited an increase in the fatty acid oxidation rate compared with cells in normal medium, and miR-33b overexpression reduced OCM-promoted fatty acid oxidation in ovarian cancer cells (Figure 3G). Notably, luminescence assays for ATP demonstrated that intracellular ATP levels in ovarian cancer cells were 1.2–1.7-fold higher in response to coculture with OCM, and OCM-enhanced ATP production was significantly counteracted in ovarian cancer cells overexpressing miR-33b (Figure 3H). Altogether, these findings suggest that overexpression of miR-33b impairs OCM-upregulated lipid metabolic activities in ovarian cancer cells.

### 3.4. miR-33b Directly Targets TAK1, Which Is Involved in Lipid Metabolism Reprogramming of Ovarian Cancer

To understand the underlying molecular mechanisms by which miR-33b suppresses ovarian cancer cell proliferation and metastasis, the TargetScan database was used to predict the potential downstream targets of miR-33b. Among the numerous targets, transforming growth factor-β-activated kinase 1 (TAK1), which has a crucial role in ovarian cancer development and metastasis, was chosen for further study [32,33]. To investigate the association between TAK1 expression and the survival rates of ovarian cancer patients, Kaplan–Meier survival curves were constructed using the PrognoScan (http://dna00.bio.kyutech.ac.jp/PrognoScan/, accessed on 8 May 2021) and Kaplan–Meier Plotter (ovarian cancer) (http://kmplot.com/analysis/index.php?p=service&cancer=ovar, accessed on 8 May 2021) databases. High TAK1 expression in ovarian cancer was significantly correlated with lower disease-free survival and overall survival (Appendix A). This finding indicates that TAK1 upregulation is likely involved in cancer metastasis, leading to poor survival outcomes in ovarian cancer. To assess whether miR-33b directly targets TAK1, a dual-luciferase assay was carried out with the wild-type and mutant 3′UTRs of TAK1. Forced miR-33b expression in HEK293 and ES-2 cells was found to induce a 60% decrease in the relative luciferase activity of the wild-type but not the mutant TAK1 3′-UTR, indicating that TAK1 is a direct target of miR-33b (Figure 4A). We next determined whether TAK1 is involved in miR-33b-mediated tumor-suppressive effects in ovarian cancer using western blot analysis. As shown in Figure 4B, transient transfection of HEK293 cells with miR-33b dose-dependently reduced TAK1 expression. The expression pattern of miR-33b and TAK1 was then examined in primary ovary carcinoma with a matched metastatic carcinoma tissue array using immunohistochemistry (IHC) and in situ hybridization (ISH). The results demonstrated that miR-33b was expressed at low levels in omental metastases compared with primary ovarian tumors. In contrast, TAK1 was more highly expressed in omental metastatic samples compared to primary tumor tissues (Figure 4C).

To determine whether miR-33b suppresses OCM-mediated metabolic reprogramming in ovarian cancer by targeting TAK1, The Cancer Genome Atlas Ovarian Cancer (TCGA-OV) data were analyzed. We stratified RNA-seq data of TCGA-OV into high and low TAK1 expression groups and performed GSEA. Interestingly, ovarian patients with high TAK1 expression showed a significant association with miR-33b-regulated cholesterol and lipid homeostasis pathways compared to the low TAK1 group (Figure 4D). Next, we evaluated the possible correlations between TAK1 and the expression of other genes and included Pearson’s correction. TCGA-OV results indicated that the regulation of lipid metabolism by PPARalpha was highly associated with TAK1 expression in patients (Figure 4E). To further validate this hypothesis, TAK1 was stably knocked out using a CRISPR/Cas9 system in ES-2 and MES-OV cells (Figure 4F). Notably, OCM-cocultured ES-2 and MES-OV cells showed a significant reduction in lipid droplet formation and cellular ATP production in TAK1 knockout cell clones compared with the scrambled control (Figure 4G,H). Collectively, our findings showed that miR-33b directly targets the 3′-UTR of TAK1, thereby suppressing OCM-mediated lipid metabolism in ovarian cancer cells.

### 3.5. miR-33b Overexpression Suppresses TAK1/FASN/CPT1A/NF-κB Signaling in Ovarian Cancer

Mounting evidence suggests that the TAK1/NF-κB signaling pathway promotes tumor aggressiveness and metastasis in ovarian cancer [23,32]. To investigate whether TAK1/NF-κB signaling plays a vital role in the tumor-suppressive effects of miR-33b in ovarian cancer, western blot analysis was performed using ovarian cancer cells with miR-33b overexpression or scrambled control under OCM or normal conditions. The results showed that overexpression of miR-33b yields a reduction in TAK1/NF-κB signaling in normal medium-cultured ES-2 cells compared with the scrambled control. Notably, OCM coculture significantly increased the expression levels of TAK1, P-TAK1, P-IKKα/β, and P-IκBα in ES-2 cells. However, OCM-stimulated TAK1/NF-κB activity in the scrambled control group was significantly suppressed upon overexpression of miR-33b (Figure 5A).

Fatty acid synthase (FASN) is a key metabolic enzyme regulating de novo fatty acid synthesis, which is crucial for the OCM-induced oncogenic properties of ovarian cancer in vitro and in vivo [23]. CPT1A is a rate-limiting transporter controlling fatty acid oxidation, and its overexpression upregulates β-oxidation of fatty acids and ATP levels to facilitate tumor cell proliferation [34]. Here, we identified that OCM remarkably enhanced FASN and CPT1A expression in ES-2 cells compared with coculture in 1% FBS-negative control medium, whereas the OCM-mediated upregulation of FASN and CPT1A was attenuated upon overexpression of miR-33b in ovarian cancer cells (Figure 5A). These results suggest that overexpression of miR-33b mitigates FASN and CPT1A regulation and that de novo lipogenesis and fatty acid degradation in ovarian cancer cells are inhibited as a result.

We next investigated whether depletion of TAK1 suppresses fatty acid biosynthesis and ATP production in OCM-cocultured ovarian cancer cells by targeting FASN and CPT1A. To this end, Spearman’s correlation analysis was conducted using the TCGA-OV dataset. Consistent with the above findings in Figure 4G,H, scatter plots indicated that high expression level of FASN and CPT1A were significantly associated with an increased level of TAK1 among ovarian cancer patients (*p* < 0.001), further confirming the presence of an intimate relation among TAK1, FASN and CPT1A in regulating cellular lipid metabolism. (Figure 5B). Our previous studies documented that the TAK1/NF-κB signaling pathway promotes tumor aggressiveness and metastasis in ovarian cancer [23]. Consistently, western blot analysis suggested that knockout of TAK1 strongly reduced OCM-initiated activation of NF-κB signaling activities in ES-2 and MES-OV cells, confirming the importance of TAK1/NF-κB signaling for OCM-induced oncogenic properties in ovarian cancer (Figure 5C). Intriguingly, TAK1^−/−^ ovarian cancer cells exhibited a marked reduction in FASN and CPT1A expression after OCM maintenance (Figure 5C). These data reveal that inhibition of TAK1 targets FASN and CPT1A and, as a consequence, alters lipid metabolism in ovarian cancer cells.

To further examine whether FASN and CPT1A are involved in oncogenic effects mediated by the NF-κB pathway in OCM-treated ovarian cancer, orlistat (FASN inhibitor) and etomoxir (CPT1A inhibitor) were employed to block the respective function of FASN and CTP1A in ovarian cancer cells. Western blot analysis observed that orlistat (30 µM) dramatically inhibited the expression of FASN in ES-2 and MES-OV cells with OCM treatment. Of note, the addition of orlistat was found to impede the phosphorylation of IKKα/β in ovarian cancer cells cultured with OCM (Figure 5D). Similarly, inhibition of CPT1A by etomoxir (40 µM) reduced P-IKKα/β expression in ES-2 and MES-OV cells maintained in OCM (Figure 5E). In summary, miR-33b acts as a tumor suppressor, inhibiting oncogenesis and targeting ovarian cancer lipid metabolism by downregulating TAK1/FASN/CPT1A/NF-κB signaling.

### 3.6. Induced Expression of miR-33b Suppresses Ovarian Cancer Growth In Vivo

To examine the effects of miR-33b on ovarian tumor dissemination in vivo, GFP-labeled ES-2 cells overexpressing miR-33b or scrambled control were intraperitoneally inoculated into SCID mice. After 3 weeks with palpable tumors observed, the mice were sacrificed. In vivo results indicated that overexpression of miR-33b led to a significant reduction in tumor mass and the number of tumor nodules. Given that the body weight was similar for both groups, the weight loss of the control group was attributable to tumor burden when compared with the miR-33b overexpressing group (Figure 6A–C). Malignant ascites accumulation is one of the typical hallmarks of ovarian cancer progression [35]. The relative ascites volume was approximately 100-fold higher in SCID mice grafted with ES-2 parental control cells, whereas overexpression of miR-33b resulted in a near absence of ascites formed in vivo (Figure 6D). The colocalization of tumor nodules and organs in the peritoneal cavity was then imaged via fluorescence microscopy. Figure 6E depicts that overexpression of miR-33b led to much less metastatic tumor seeding in distant organs of mice. Subsequently, IHC was performed to assess the effect of miR-33b on the TAK1/FASN/CPT1A/NF-κB axis and Ki67 expression in vivo. The results showed that miR-33b overexpression reduced TAK1, FASN, CPT1A, P-IKKα/β, P-IκBα, and Ki67 protein expression in xenograft tumor tissues (Figure 6F). These results suggest that downregulation of miR-33b could trigger oncogenic properties through activation of the TAK1/FASN/CPT1A/NF-κB signaling axis and, as a consequence, facilitate peritoneal metastases of ovarian cancer.

## 4. Discussion

Emerging evidence indicates that the TME plays a critical role in shaping the epigenetic states of cancer cells [36]. DNA hypermethylation of tumor-suppressive miRNAs leads to miRNA silencing and therefore contributes to tumor development and progression [37]. In this study, we identified that miR-33b is downregulated by epigenetic silencing in metastatic ovarian tumor tissues in the omentum compared to primary ovarian tumor tissues. miR-33b has been demonstrated to exert tumor-suppressing effects by impairing OCM-promoted oncogenic properties, tumor growth, and peritoneal metastases in vitro and in vivo. Proteomic profiling revealed the importance of miR-33b in regulating metabolic pathways in ovarian cancer cells by suppressing their lipid metabolic activities, such as de novo lipogenesis and fatty acid oxidation when cocultured in OCM. Mechanistic studies emphasized that miR-33b directly targets TAK1, which leads to downregulation of FASN and CPT1A, explaining the inhibitory effects of miR-33b on the lipid metabolic activities of ovarian cancer cells. Importantly, FASN or CPT1A suppression results in a reduction in NF-κB activities and, as a consequence, affects the proliferation, migration, and invasion of ovarian cancer cells. These findings suggest that a lipid-rich microenvironment leads to methylation-induced silencing of miR-33b in ovarian cancer cells, enabling adaptation and metastatic progression. Therefore, targeting the miR-33b-mediated pathway may be a promising therapeutic approach for preventing ovarian cancer peritoneal metastases.

Emerging evidence has indicated that cancer cells can adapt to environmental conditions through epigenetic mechanisms in response to their niche during tumor progression [38]. Epigenetic changes, including DNA methylation and miRNA-mediated silencing, primarily drive tumorigenesis and metastasis in multiple cancer types, including ovarian cancer [39,40,41]. In ovarian cancer, tumor suppressor genes or miRNAs are often silenced by epigenetic events, consequently promoting cancer proliferation, metastasis, and chemoresistance [24,42]. The omental microenvironment is a major metastatic site and plays a key role in ovarian cancer peritoneal metastases [39,43]. Recent studies have reported aberrant epigenetic crosstalk between ovarian cancer cells and adipocytes in the omental metastatic TME, which contributes to the metastatic progression of ovarian cancer [5,22]. However, the epigenetic mechanisms underlying the adaptation of ovarian cancer cells to the omental TME remain largely unknown. In the present study, miR-33b was found to be frequently hypermethylated in human omental metastatic tissues compared to corresponding primary ovarian tumors. miR-33b is a key regulator of fatty acid oxidation and cholesterol metabolism; it is an intronic miRNA located within sterol regulatory element-binding protein 1 (SREBF1), an essential transcriptional modulator of lipid metabolism [44]. Previous evidence indicates that overexpression of miR-33b can reduce fatty acid degradation by decreasing the activities of enzymes such as CPT1A in hepatic cell lines [28]. Notably, we discovered that miR-33b expression was low in omental metastases compared with primary tumors. In line with observations in clinical samples, miR-33b was expressed at a low level in various ovarian cancer cell lines compared with HOSE cells. However, treatment with the demethylation agent 5-Aza-dC restored the expression of miR-33b in ovarian cancer cells. MS-PCR and pyrosequencing analysis confirmed the presence of methylated CpG marks on the promoter region of the miR-33b gene in ovarian cancer cells and clinical samples. These data suggest that downregulation of miR-33b in ovarian cancer is associated with DNA hypermethylation and, as a result, promotes ovarian tumor development and metastases.

Approximately 80% of EOC patients present peritoneal dissemination and omentum infiltration [45]. The omentum is a large abdominal organ rich in adipocytes and works as a preferential metastatic site for ovarian cancer [46]. In ovarian cancer, the omental microenvironment facilitates metastatic dissemination of cancer cells, which is associated with a poor prognosis [23]. Emerging evidence suggests that metastatic ovarian cancer cells utilize fatty acids from adipocytes in the omentum to fuel mitochondrial ATP production for tumor growth and aggressiveness [23,46]. Coculture of ovarian cancer cells with omental adipocytes revealed that ovarian cancer cells promote lipolysis in adipocytes, transfer free fatty acids to tumor cells, activate fatty acid oxidation, and enhance cell growth [47,48]. Moreover, previous studies have developed OCM as a model to investigate interactions between the omental tumor microenvironment in the peritoneal cavity and the oncogenic capacities of metastatic ovarian cancer cells [33]. Intriguingly, ovarian cancer cells were observed to undergo metabolic reprogramming and generate ATP from fatty acid oxidation instead of aerobic glycolysis when cultured in OCM [23]. Indeed, reprogramming of lipid metabolism plays essential roles in providing energy, lipid-mediated signaling, and the formation of a premetastatic niche in ovarian cancer [49]. Studies on ovarian cancer have suggested that crucial enzymes that mediate metabolic reprogramming, such as FASN and CPT1A, are overexpressed in tumor tissues, which is associated with poor prognosis and survival rates [23,34]. Consistent with previous findings, our results revealed that OCM stimulates ovarian cancer cells to use free fatty acids from the omental environment for de novo lipogenesis and fatty acid β-oxidation, thereby supporting ovarian cancer cell proliferation and migration/invasion. miR-33b is known to regulate fatty acid metabolism and to target lipid metabolism genes [28]. Here, we found that miR-33b significantly mitigated the elevated lipogenesis, lipolysis and β-oxidation activities of ovarian cancer cells in fatty acid-enriched OCM; accordingly, miR-33b inhibited OCM-mediated ovarian cancer cell proliferation and migration/invasion abilities in vitro. As a highly metastatic ovarian cancer cell line, ES-2 cells has been employed to establish in vivo model of peritoneal metastases and it is shown that miR-33b suppressed ovarian tumor growth and dissemination—of animal model. Mechanistically, miR-33b directly targets TAK1, thereby decreasing the expression of FASN and CPT1A in ovarian cancer cells, which reduces OCM-promoted fatty acid synthesis and ATP production. Ultimately, activation of the NFκB signaling pathway is repressed, and as a result, the aggressiveness and oncogenic properties of ovarian cancer cells enhanced by OCM are impaired.

TAK1/NF-κB signaling has been implicated in the aggressiveness of ovarian cancer in OCM [23,32]. Consistent with previous reports, we found that OCM induces significant activation of the TAK1/NFκB pathway in ovarian cancer cells. Aside from working as a crucial regulator of the NFκB signaling pathway, TAK1 also plays a pivotal role in lipid metabolism and homeostasis. TAK1 deficiency has been shown to promote apoptosis, reduce adipocyte number and facilitate browning of white adipocyte tissue (WAT) in mice fed a high-fat diet [50]. Adipocyte-specific loss of TAK1 diminished the levels of phosphorylated IκBα, a key marker of activation of the canonical NF-κB pathway, which may contribute to cell death in epididymal WAT in TAK1 knockout mice [51]. Recently, an increasing number of studies have investigated the role of TAK1 in hepatic lipid metabolism. Sayaka et al. found that hepatic TAK1 deletion resulted in a significant reduction in the fatty acid oxidation rate and *CPT1A* transcriptional level in a mouse model [52]. Meanwhile, Dan et al. reported that treatment with the TAK1 inhibitor 5Z-7-ox attenuated lipid accumulation and FASN expression induced by PAOA in wild-type and Tnip3-deficient primary hepatocytes [53]. The NFκB pathway is frequently activated in various human malignancies and plays an important role in the tumor cell response to stress, such as alterations in lipid metabolism [54,55]. Shogo et al. noted that knockdown of FABP5 induced a decrease in intracellular fatty acid levels; downregulation of genes involved in lipolysis, β-oxidation, and lipogenesis; and suppression of NFκB activity in the PC-3 prostate cancer cell line and M231 triple-negative breast cancer cell line [56]. It has been demonstrated that de novo lipogenesis and lipid desaturation are metabolic markers and contribute to maintaining ovarian cancer cell stemness [57]. Intriguingly, de novo lipogenesis and subsequent lipid desaturation dramatically promoted the activity of the NFκB signaling pathway and ovarian cancer stem cells [57,58]. Our present study observed that depletion of TAK1 leads to decreased FASN and CPT1A expression, and consequently, this inhibition of lipid metabolic activities results in suppression of NF-κB activation in OCM cocultured ovarian cancer cells.

## 5. Conclusions

Given the poor prognosis and survival outcomes of HGSOC patients, the investigation of effective therapeutic interventions for ovarian cancer with peritoneal metastases is of the utmost urgency. Tumor suppressor miRNAs have emerged as promising targets for cancer therapy because miRNAs have been shown to regulate thousands of potential target genes and to control important cellular events in tumor development and progression [59,60]. In this study, the ISH and IHC results showed an inverse correlation between miR-33b and TAK1 in primary and metastatic ovarian cancer tissues. TAK1 has been confirmed to enhance the oncogenesis and aggressiveness of ovarian cancer cells, and analysis of TCGA and other databases revealed that TAK1 expression in tumors is negatively correlated with ovarian cancer patient survival. Of note, the in vivo mouse study indicated that ectopic expression of miR-33b reduced tumor mass, ascites volume, and the number of tumor nodules. Conclusively, we observed the clinical relevance of miR-33b, and our findings suggest that miR-33b might be an attractive therapeutic strategy for the management of metastatic ovarian cancer with peritoneal dissemination.

## Figures and Tables

**Figure 1 cancers-13-04795-f001:**
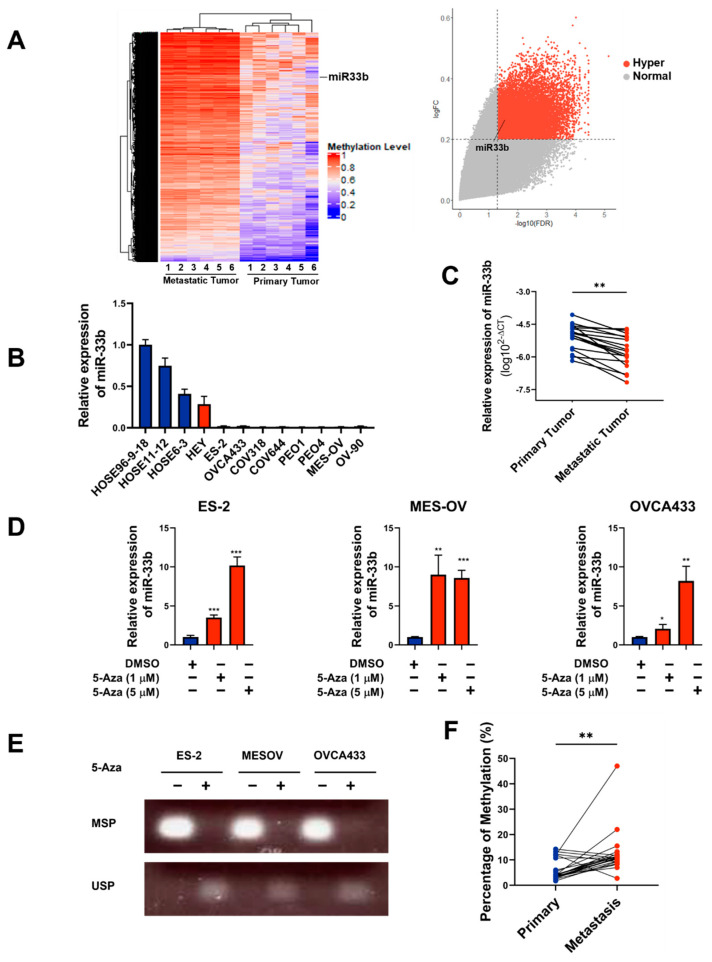
DNA methylation is associated with the downregulation of miR-33b in ovarian cancer. (**A**) DNA methylation was detected in six pairs of primary and metastatic tumors from ovarian cancer patients. Right panel: global DNA methylation. Left panel: significant differences in methylation level of miR-33b host gene. (**B**) qPCR analysis of miR-33b expression in human ovarian surface epithelial (HOSE) cell lines and ovarian cancer cell lines. U6 was used as an internal control. (**C**) MiR-33b expression level was measured in pairs of primary and metastatic ovarian tumors (*n* = 18 pairs). (**D**) In ovarian cancer cells, the expression of miR-33b was restored upon treatment with 5-Aza-dc (1 μM or 5 μM) for 96 h. (**E**) DNA methylation of miR-33b promoter was assessed by MS-PCR in ovarian cancer cells upon 5-Aza-dC (5 μM) treatment for 96 h. (**F**) Pyrosequencing analysis detected methylation percentage of miR-33b promoter in pairs of primary tumor tissues and omental specimens (*n* = 22 pairs). Remarkable differences were observed in 18 pairs of ovarian cancer samples. Meanwhile, no significant difference was shown in the other 4 pairs of clinical tumor tissues. (* *p* < 0.05, ** *p* < 0.01, *** *p* < 0.001).

**Figure 2 cancers-13-04795-f002:**
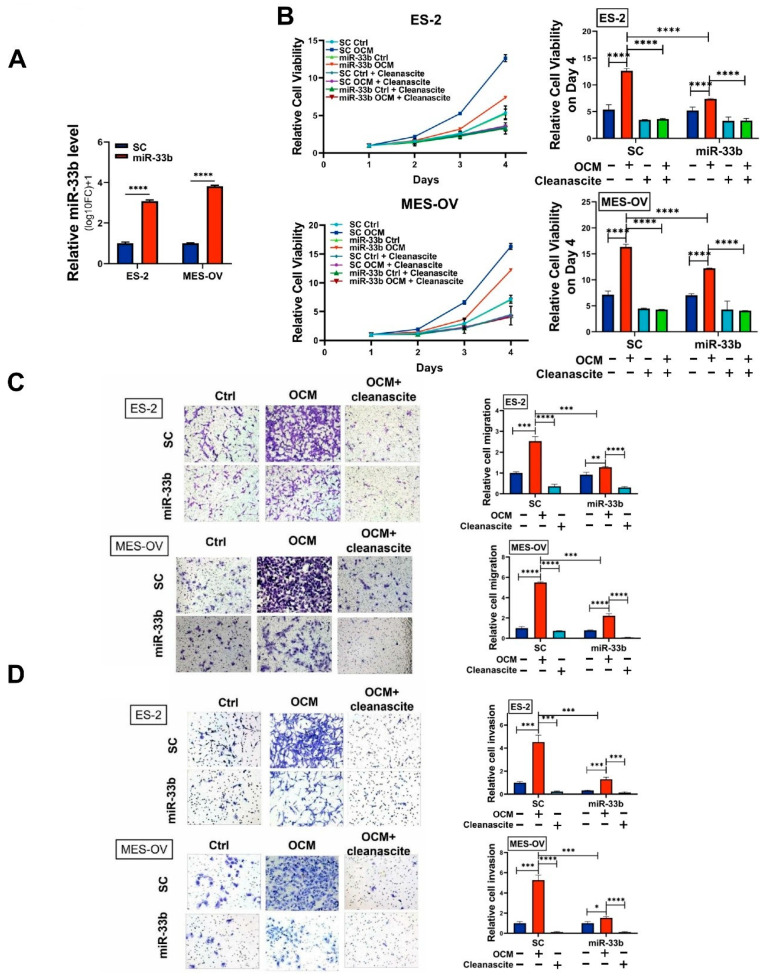
miR-33b abrogates oncogenic properties induced by OCM in ovarian cancer cells. (**A**) Taqman microRNA assay validated the overexpression of miR-33b in ES-2 and MES-OV cells. U6 was used as an internal control. (**B**–**D**) Ovarian cancer cell proliferation, migration, and invasion of stable miR-33b overexpressing or scrambled control ES-2 and MES-OV cells were examined under the treatment of control medium, omental conditioned medium (OCM) or OCM with Cleanascite (24 h for transwell migration and invasion assay). Cleanascite was applied to remove the lipids of the culture medium. Error bars denoted mean ± SD (*n* = 3). (* *p* < 0.05, ** *p* < 0.01, *** *p* < 0.001, **** *p* < 0.0001). Magnification: 40×.

**Figure 3 cancers-13-04795-f003:**
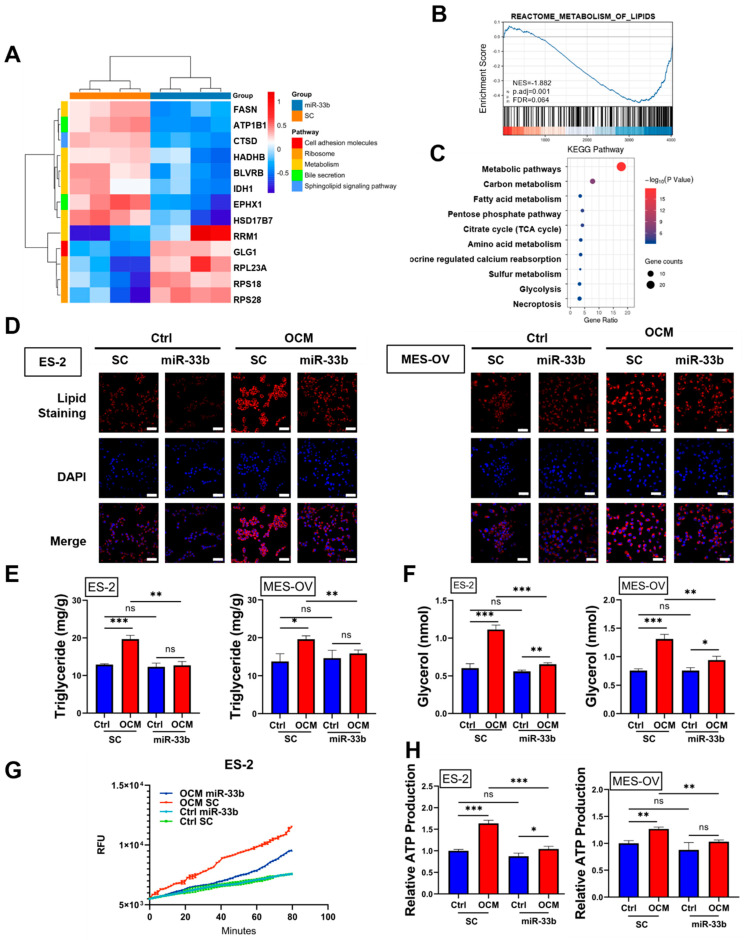
miR-33b regulates OCM-mediated lipid metabolic reprogramming of ovarian cancer cells. (**A**) Stable polyclonal ES-2 cell line with overexpression of miR-33b or scrambled control was treated with OCM for 24 h, and subsequently, total protein was extracted for LC-MS/MS proteomic study. Some of the downregulated and upregulated genes in stable miR-33b overexpressing ES-2 cells relative to scrambled control ES-2 cells were presented by heatmap. (**B**) Gene set enrichment analysis (GSEA) was employed to show the biological pathways that enriched in ES-2 cells overexpressing miR-33b compared to scrambled control. (**C**) Functional enrichment analysis was performed on 92 genes downregulated by miR-33b overexpression using KEGG database. (**D**–**H**) Ovarian cancer cell lines (ES-2 and MES-OV) were treated as described in (**A**), and lipid metabolic activities were examined. (**D**) Lipid droplet formation was assessed by immunofluorescent and lipid staining analyses. Lipid droplet was counterstained by Nile red (in red) and nuclei were stained by DAPI (in blue) before visualization with Zeiss LSM 980 (Zeiss, Jena, Germany). Scale bar: 50 μm. (**E**). The triglyceride level of ovarian cancer cells was quantified by Triglyceride Assay. (**F**) Lipolysis activity of ES-2 and MES-OV cells was detected by Lipolysis Colorimetric Assay. (**G**) β-oxidation of fatty acid was continuously monitored by Fatty Acid Oxidation Assay over time. (**H**) Cellular ATP production of ovarian cancer cells was determined by Luminescent ATP Detection Assay. (ns, no significant difference, * *p* < 0.05, ** *p* < 0.01, *** *p* < 0.001).

**Figure 4 cancers-13-04795-f004:**
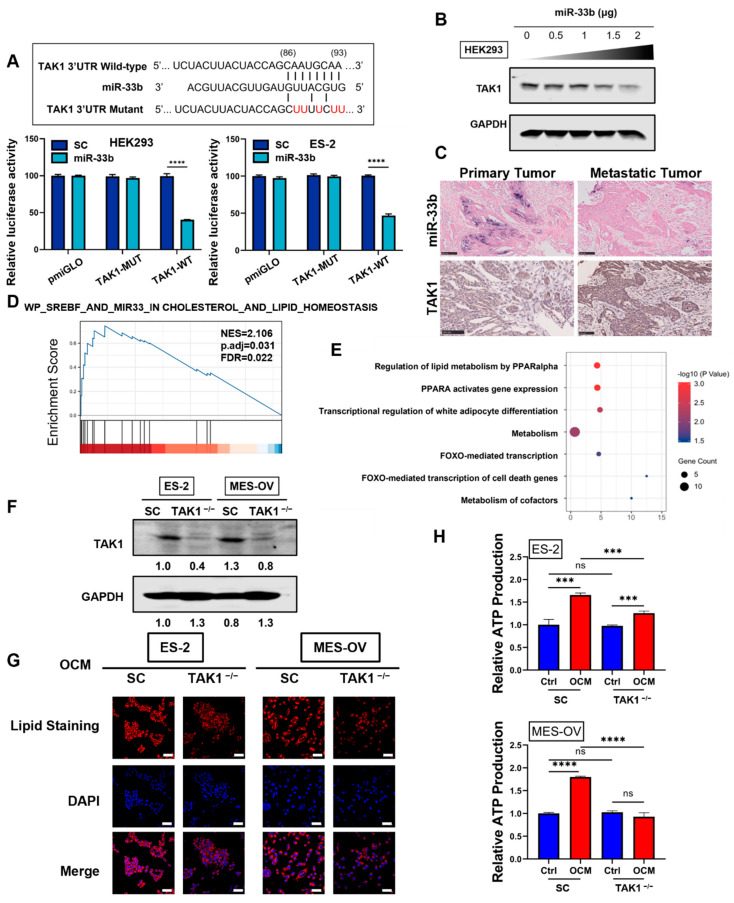
TAK1, a direct target of miR-33b, is associated with lipid metabolism of ovarian cancer cells. (**A**) Schematic representation of putative miR-33b binding site in wild-type or mutant TAK1 3′UTR (upper panel). Luciferase reporter activity in HEK293 or ES-2 cells co-transfected with wild-type or mutant 3′UTR TAK1 constructs with pCMV-miR33b plasmid or scrambled control was determined by the Dual-Luciferase^®^ Reporter Assay System (lower panel). (**B**) HEK293 cells were transfected with different amounts of pCMV-miR33b plasmid for 48 h, and TAK1 expression was measured using western blot analysis. (**C**) Representative micrographs depicting the primary ovary carcinoma with matched metastatic carcinoma tissue array (OV812) stained for miR-33b by in-situ hybridization (ISH) and TAK1 by immunohistochemistry (IHC); scale bar: 100 μm. (**D**) WikiPathways regulated by TAK1 in TCGA-OV were explored by Gene Set Enrichment Analysis (GSEA) analysis. (**E**) Genes highly linked to TAK1 in TCGA-OV were enriched by searching the Reactome database. (**F**) TAK1 knockout in ES-2 and MES-OV cells was confirmed by western blotting assay. (**G**) A lipid staining assay visualized the de novo lipogenesis of TAK1 knockout or scrambled control ES-2 and MES-OV cells upon 24 h OCM coculture. Scale bar: 50 μm. (**H**) ATP level in TAK1 knockout or scrambled control ES-2 and MES-OV cells with OCM treatment (24 h) was detected by ATP detection assay. *** *p* < 0.001, **** *p* < 0.0001). The original western blot images were shown in the Appendix A.

**Figure 5 cancers-13-04795-f005:**
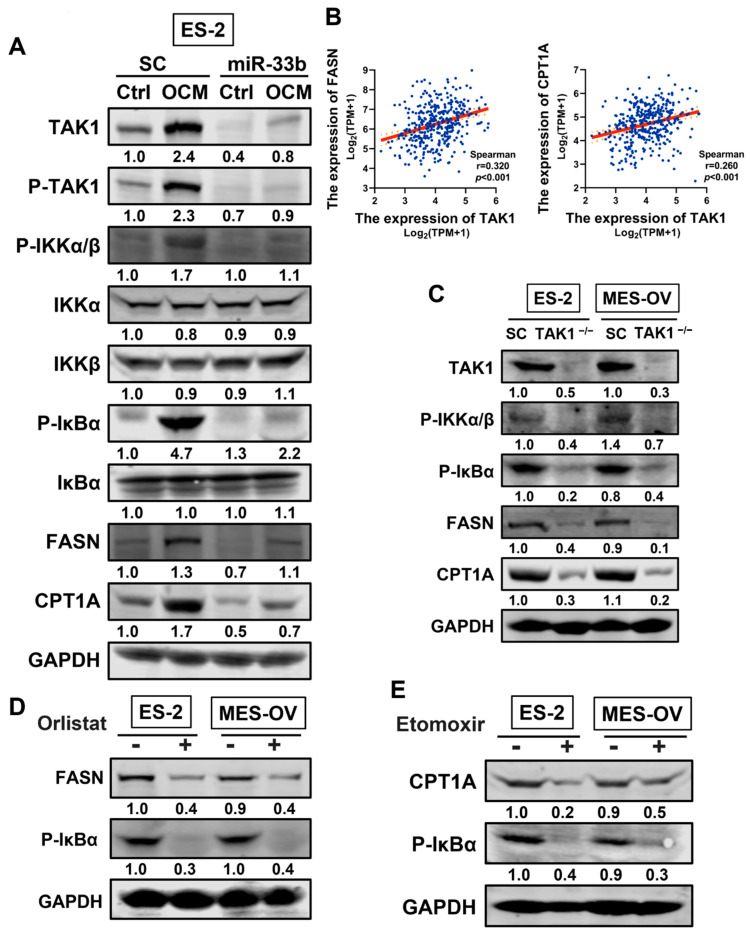
miR-33b targeting TAK1/FASN/CPT1A/NF-κB signaling axis in ovarian cancer cells. (**A**) Scrambled control or stable miR-33b overexpressing ES-2 and MES-OV cells were treated with OCM or 1% FBS normal medium for 24 h. Protein contents of TAK1, P-TAK1, P-IKKα/β, IKKα, IKKβ, P-IκBα, IκBα, FASN, CPT1A, GAPDH were examined by WB. (**B**) Scatter plots depicting the correlation between TAK1 and FASN (right panel) or CPT1A (left panel) in TCGA-OV dataset. (**C**) ES-2 and MES-OV cells with scrambled control or stable knockout of TAK1 were cultured with OCM for 24 h. Protein contents of TAK1, P-IKKα/β, P-IκBα, FASN, CPT1A, GAPDH were determined by WB. (**D**) OCM co-cultured ovarian cancer cells were subjected to co-treatment of FASN inhibitor orlistat (30 µM) or DMSO for 24 h. Protein contents of FASN, P-IκBα, GAPDH were confirmed by WB. (**E**) OCM-cultured ovarian cancer cells were subjected to co-treatment of CPT1A inhibitor etomoxir (40 µM) or DMSO for 24 h. Protein contents of CPT1A, P-IκBα, GAPDH were evaluated by WB. The original western blot images were shown in the Appendix A.

**Figure 6 cancers-13-04795-f006:**
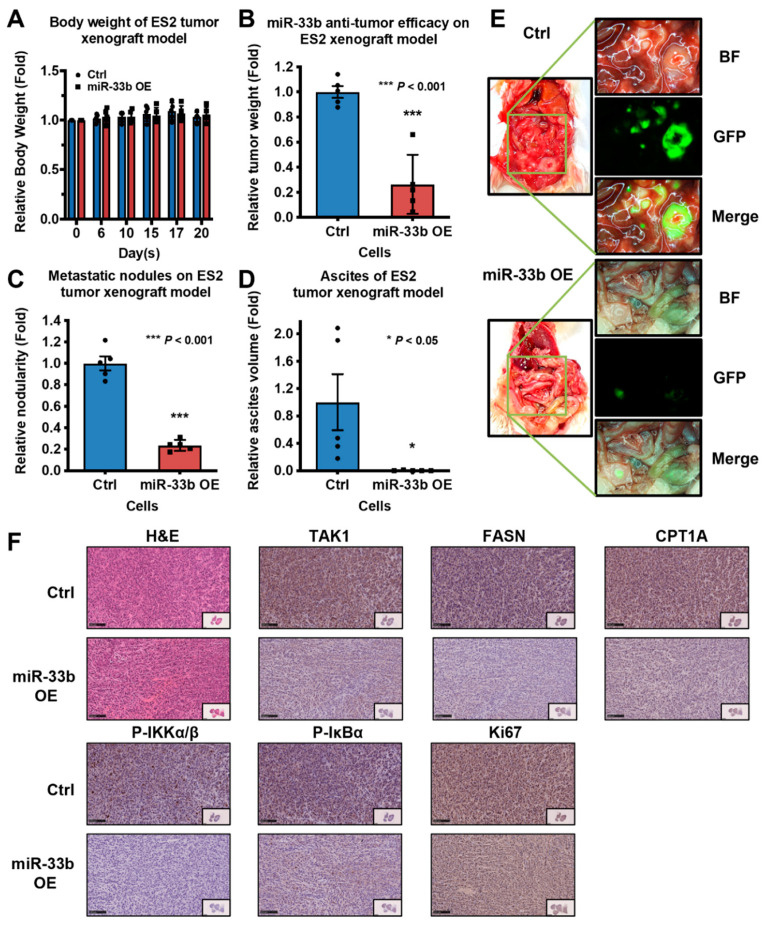
Overexpression of miR-33b inhibits tumor dissemination of metastatic ovarian cancer cells in vivo. (**A**–**D**) miR-33b overexpressing or scrambled control ES-2 cells with GFP label were intraperitoneally injected into 6–8 weeks old female SCID mice in a group of 5. The bodyweight of the mice was continuously monitored. The mice were sacrificed about 3 weeks later, and the body weight, tumor mass, number of tumor nodules, and ascites volume were measured. OE: overexpression. (**E**) GFP-labelled ES-2 cells were disseminated in the abdominal cavity, and tumor nodules were formed, which were visualized by fluorescence stereomicroscopy. BF: bright field. (**F**) The expression of TAK1, FASN, CPT1A, P-IKKα/β, P-IκBα, Ki67 in tumor tissues of miR-33b overexpression and scrambled control groups were measured by IHC. scale bar: 100 μm.

## Data Availability

The protein expression profile of miR-33b overexpressing ES-2 cells upon OCM treatment has been deposited in PRIDE (Project accession: PXD026901). The dataset of methylome profiling generated and analyzed during the current study is not publicly available because other unpublished studies are using but are available from the corresponding author on reasonable request. All other data generated or analyzed during this study are included in this published article.

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
