# Peer review of "Epigenetic Silencing of miR-33b Promotes Peritoneal Metastases of Ovarian Cancer by Modulating the TAK1/FASN/CPT1A/NF-κB Axis"

_cancers, 2021, doi:10.3390/cancers13194795_

Round 1
Reviewer 1 Report
Overall, this is an interesting study that demonstrates a role for miR-33b in ovarian cancer metastasis that provides underlying mechanistic insights. The authors have included both primary patient samples and cell lines, and performed both in vitro and in vivo studies. The manuscript is well-written. A number points of feedback follow that would make the manuscript even stronger:
Major comments:
- Why was the clear cell line ES-2 selected for the in vivo studies rather than one of the many HGSC cell lines used earlier in the studies? Please both explain the rationale and acknowledge this limitation in the discussion, as it potentially limits extending results to HGSC.
- Figure 5B – Although the left panel (TAK1/FASN) is statistically significant, it does not pass the “eyeball test.” Same concern regarding the right panel, which in the text is accurately described as positively correlated (without significance). These findings should be de-emphasized in the text and/or explanations provided for why a stronger correlation for both is not seen.
- Figure 6A – I think you are showing the controls as the comparison group for the “Relative Body Weight.” Is the y-axis actually % rather than g? Additionally, if the miR-33b overexpressing animals have 1 ml of ascites (Figure 6D) and almost 1 g more tumors than the controls (Figure 6B), shouldn’t they weigh more? If the animal weights are the same as controls, this suggests the miR-33b overexpressing animals are losing weight as a result of their cancer burden. Please address in the text. Changing the graph to show the weight changes of individual mice in each group over time may be clearer.
Minor comments:
- Please clarify in the methods and results sections that the stable miR-33b overexpressing lines are polyclonal.
- Figure 1F – Can you please further label the two panels of the figure and update the legend to clarify what the difference is between the left and right panels? I infer the left is non-ovarian cancers and the right is ovarian cancers, but it’s not totally clear by just looking at the figure and legend.
- Page 8, line 317 – There’s an extra “T” at the start of the sentence.
- Figure 2 – Can you please add what Cleanascite is to the figure legend? And also the OCM abbreviation? Will help readers follow.
Author Response
Major Comments:
- Why was the clear cell line ES-2 selected for the in vivo studies rather than one of the many HGSC cell lines used earlier in the studies? Please both explain the rationale and acknowledge this limitation in the discussion, as it potentially limits extending results to HGSC
Response: Thanks for your valuable suggestions, ES-2 ovarian cancer cell line is highly metastatic and a large number of publications employed the ES-2 cell model to study tumor dissemination. In fact, apart from HGSOC, ovarian clear cell subtype carcinoma is also commonly observed in omental metastasis.
- Figure 5B – Although the left panel (TAK1/FASN) is statistically significant, it does not pass the “eyeball test.” Same concern regarding the right panel, which in the text is accurately described as positively correlated (without significance). These findings should be de-emphasized in the text and/or explanations provided for why a stronger correlation for both is not seen.
Response: Figure 5B has been revised using “TPM” instead of “FPKM” so that both FASN and CPT1A are significantly correlated with TAK1 expression in the TCGA-OV dataset.
- Figure 6A – I think you are showing the controls as the comparison group for the “Relative Body Weight.” Is the y-axis actually % rather than g? Additionally, if the miR-33b overexpressing animals have 1 ml of ascites (Figure 6D) and almost 1 g more tumors than the controls (Figure 6B), shouldn’t they weigh more? If the animal weights are the same as controls, this suggests the miR-33b overexpressing animals are losing weight as a result of their cancer burden. Please address in the text. Changing the graph to show the weight changes of individual mice in each group over time may be clearer.
Response: (1) Figure 6 has been revised. The fold is presented in the y-axis. (2) As compared to the control group, the miR-33b overexpressing group exhibited lower tumor weight and less ascites volume. Given that both groups showed similar body weight, the mice from the control group may be losing weight as a result of tumor burden. (3) Figure 6A has been updated that showing the weight changes of individual mice in each group over time.
Minor Comments:
- Please clarify in the methods and results sections that the stable miR-33b overexpressing lines are polyclonal.
Response: It has been revised as “stable polyclonal miR-33b-overexpressing cell line” in the method and result part.
- Figure 1F – Can you please further label the two panels of the figure and update the legend to clarify what the difference is between the left and right panels? I infer the left is non-ovarian cancers and the right is ovarian cancers, but it’s not totally clear by just looking at the figure and legend.
Response: The right and left panels have been merged as one panel in figure 1F.
- Page 8, line 317 – There’s an extra “T” at the start of the sentence.
Response: The extra “T” has been removed.
- Figure 2 – Can you please add what Cleanascite is to the figure legend? And also the OCM abbreviation? Will help readers follow.
Response: It has been revised as “Cleanascite was applied to remove the lipids of culture medium” and “OCM (omental conditioned medium)” in the figure legend.
Reviewer 2 Report
Xueyu Wang et al., evaluate the epigenetic role of miR-33b in ovarian cancer metastasis by modulating the lipid metabolic activities via TAK1/FASN/CPT1A/NF-κb axis. It is impressive clinical research data to find out the cross talk between omental tumor microenvironment and cancer cells to initiate the metastatic invasion.
Authors use validated global hyper methylated and normal to compare the CHIP assay data and well explained all the experimental details too.
Here is some of the comments.
- The basic point/question I want know that, the authors used omental conditioned medium (OCM) which is made-up of several cells and factors including normal fibroblasts, cancer associated fibroblast, mesothelial cells and ECM protein. Out of these, which are the responsible factor to induce the epigenetic silencing of miR-33b.
- Page 7, Figure 1A is missing or not labeled properly.
- Since miR-33b involved in proliferation and migration, what is the role of its target TAK1 (after its silencing) in proliferation and migration.
- Type error on line 163 “OCMco--cultured ovarian”.
Author Response
Comments:
1.The basic point/question I want know that, the authors used omental conditioned medium (OCM) which is made-up of several cells and factors including normal fibroblasts, cancer associated fibroblast, mesothelial cells and ECM protein. Out of these, which are the responsible factor to induce the epigenetic silencing of miR-33b.
Response: In this study, OCM (omental conditioned medium) was employed to mimic the lipid-rich metastatic tumor microenvironment of the intraperitoneal cavity. Indeed, omentum harbors a variety of cell types, such as cancer-associated fibroblast and mesothelial cells. Aside from these, it is believed that cancer-associated adipocytes play an important role in triggering epigenetic alterations of ovarian cancer cells. Our lab by Chen et al. have recently reported that lipid metabolism is significantly associated with OCM-cultured ovarian cancer cells based on the proteomic results [1]. In this study, it was found that ovarian cancer cells exert metabolic reprogramming in OCM. Consistently, Morrison AJ et al. demonstrated that metabolic processes in cancer cells fuel malignant growth, in part, through epigenetic regulation of gene expression programs important for proliferation and adaptive survival [2]. Therefore, lipid enriched tumor microenvironment that imitated by OCM contributes to the epigenetic changes of ovarian cancer cells, which promotes peritoneal metastases of ovarian cancer.
- Page 7, Figure 1A is missing or not labeled properly.
Response: Figure 1A has been added to the figure legend.
- Since miR-33b involved in proliferation and migration, what is the role of its target TAK1 (after its silencing) in proliferation and migration.
Response: According to publications “Elevated TAK1 augments tumor growth and metastatic capacities of ovarian cancer cells through activation of NF-κB signaling” from our lab previously [3], TAK1 is frequently upregulated in ovarian cancer. In addition, TAK1 overexpression could increase the proliferation, migration/invasion while depletion of TAK1 or TAK1 inhibitor led to a decrease of cellular proliferation in ovarian cancer.
- Since miR-33b involved in proliferation and migration, what is the role of its target TAK1 (after its silencing) in proliferation and migration.
Response: It has been revised as “OCM co-cultured ovarian”.
References:
- Chen RR, Yung MMH, Xuan Y, Zhan S, Leung LL, Liang RR, Leung THY, Yang H, Xu D, Sharma R et al: Targeting of lipid metabolism with a metabolic inhibitor cocktail eradicates peritoneal metastases in ovarian cancer cells. Commun Biol 2019, 2:281.
- Morrison AJ: Cancer cell metabolism connects epigenetic modifications to transcriptional regulation. FEBS J 2021.
- Cai PC, Shi L, Liu VW, Tang HW, Liu IJ, Leung TH, Chan KK, Yam JW, Yao KM, Ngan HY et al: Elevated TAK1 augments tumor growth and metastatic capacities of ovarian cancer cells through activation of NF-kappaB signaling. Oncotarget 2014, 5(17):7549-7562.